# Effect of Camel Milk on Glucose Homeostasis in Patients with Diabetes: A Systematic Review and Meta-Analysis of Randomized Controlled Trials

**DOI:** 10.3390/nu14061245

**Published:** 2022-03-15

**Authors:** Refat AlKurd, Nivine Hanash, Narmin Khalid, Dana N. Abdelrahim, Moien A. B. Khan, Lana Mahrous, Hadia Radwan, Farah Naja, Mohamed Madkour, Khaled Obaideen, Katia Abu Shihab, MoezAlIslam Faris

**Affiliations:** 1Department of Nutrition, Faculty of Pharmacy and Medical Sciences, University of Petra, P.O. Box 961343, Amman 11196, Jordan; ralkurd@uop.edu.jo; 2Care and Public Health Research Institute (CAPHRI), Maastricht University, 6211 LM Maastricht, The Netherlands; hanachnivine@gmail.com; 3Department of Clinical Nutrition and Dietetics, University of Sharjah, Sharjah P.O. Box 27272, United Arab Emirates; u17100078@sharjah.ac.ae (N.K.); hradwan@sharjah.ac.ae (H.R.); fnaja@sharjah.ac.ae (F.N.); u17100694@sharjah.ac.ae (K.A.S.); 4Department of Nutrition and Dietetics, Bahrain Defense Force Royal Medical Services Hospital, Riffa P.O. Box 28743, Bahrain; 5Clinical Nutrition and Dietetics, Faculty of Pharmacy, Applied Science Private University, Amman 11931, Jordan; danaabdelrahim@gmail.com; 6Nutrition Studies Research Group, Department of Family Medicine, College of Medicine and Health Sciences, United Arab Emirates University, Al-Ain P.O. Box 15551, United Arab Emirates; moien.khan@uaeu.ac.ae; 7Primary Care, NHS North West London, London TW3 3EB, UK; 8Department of Health Sciences/Track of Clinical Nutrition, College of Health and Rehabilitation, Princess Nourah Bint Abdulrahman University, Riyadh 12461, Saudi Arabia; lana.mahrous@gmail.com; 9Department of Medical Laboratory Sciences, University of Sharjah, Sharjah P.O. Box 27272, United Arab Emirates; mmadkour@sharjah.ac.ae; 10Sustainable Energy & Power Systems Research Centre, RISE, University of Sharjah, Sharjah P.O. Box 27272, United Arab Emirates; khaled.obaideen@gmail.com

**Keywords:** Arabian *camel*, *Camelus dromedaries*, complementary and alternative medicine (CAM), glucometabolic parameters, glycemic control, insulin resistance

## Abstract

The effects of camel milk (CM) intake on glycemic control in patients with diabetes are controversial. This systematic review and meta-analysis of randomized controlled trials (RCTs) was conducted to summarize the effect of CM intake on glucose homeostasis parameters in patients with both types of diabetes mellitus; T1DM and T2DM. We searched Google Scholar, PubMed/MEDLINE, EBSCO host, CINAHL, ScienceDirect, Cochrane, ProQuest Medical, Web of Science, and Scopus databases from inception until the end of November 2021. Relevant RCTs were identified, and the effect size was reported as mean difference (MD) and standard deviation (SD). Parameters of glycosylated hemoglobin (HbA1c), fasting blood glucose (FBG), postprandial blood glucose (PBG), fasting serum insulin (FI), insulin resistance (expressed in terms of HOMA-IR), insulin dose (ID) received, serum insulin antibody (IA), and C-peptide (CP) were tested. Out of 4054 collected articles, 14 RCTs (total 663 subjects) were eligible for inclusion. The pooled results obtained using a random-effects model showed a statistically significant decrease in HbA1c levels (MD, −1.24, 95% confidence interval (CI): −2.00, −0.48, *p* < 0.001 heterogeneity (*I*^2^) = 94%) and ID received (MD, −16.72, 95% CI: −22.09, −11.35 *p* < 0.00001, *I*^2^ = 90%), with a clear tendency was shown, but non-significant, to decrease FBG (MD, −23.32, 95% CI: −47.33, 0.70, *p* = 0.06, *I*^2^ = 98%) in patients with diabetes who consumed CM in comparison to those on usual care. Conversely, the consumption of CM did not show significant reductions in the rest of the glucose homeostasis parameters. Subgroup analysis revealed that patients with T2DM were more beneficially affected by CM intake than those with T1DM in lowering FBG, while patients with T1DM were more beneficially affected by CM intake than those with T2DM in lowering HbA1c. Both fresh and treated (pasteurized/fermented) CM gave similar beneficial effects in lowering HbA1c. Lastly, a relatively superior effect for longer duration on shorter duration (>6 months, ≤6 months, respectively) of CM intake is found in lowering HbA1c. To conclude, long-term consumption of CM by patients with diabetes could be a useful adjuvant therapy alongside classical medications, especially in lowering the required insulin dose and HbA1c. Due to the high heterogeneity observed in the included studies, more controlled trials with a larger sample size are warranted to confirm our results and to control some confounders and interfering factors existing in the analyzed articles.

## 1. Introduction

Diabetes mellitus (DM) is one of the most prevalent chronic diseases in the world, with significant health, psychological, social, and economic consequences [1]. Globally, an estimate of one in ten adults are living with diabetes, with a total incidence of 537 million (20–79 years) in 2021, and a rise of 16% (74 million) as per the 2019 figures of the International Diabetes Federation (IDF) [2].

Diabetes management through hypoglycemic agents, insulin, and diet therapy are very well investigated. Over the years, many types of traditional food treatments have been used to treat patients with diabetes and control disease complications [3,4]. Recently, complementary and alternative medicine (CAM) has been extensively studied in diabetes management with different types and modalities that have been proposed to help counteract the adverse macrovascular and microvascular complications of the disease with the least economic impact [5]. Patients with type 2 diabetes (T2DM) are driven to manage the complexities of their condition, enhance their health, and ease complications through the use of CAM [6]. The prevalence of using CAM therapies among T2DM patients has ranged from 17 up to 73% [7,8].

Among these, dromedary camel (*Camelus dromedaries*, also known as an Arabian *camel*) milk (CM) has been gaining considerable attention from scientific and medical communities. Owing to its unique composition characterized by the presence of a plethora of essential nutrients [9,10] and bioactive constituents [11,12], CM has been shown to have various beneficial effects on human health and nutrition, as well as many functional properties related to the prevention and management of many chronic and acute diseases [10,11,13,14,15,16,17,18,19,20]. One of the most compelling effects is the anti-diabetic and glucose-lowering effects of CM. A wide spectrum of published studies, including human experimental, observational, in vivo, and in vitro studies, have confirmed the promising effects of CM in this domain [21]. However, CM interventions in patients with both types of diabetes, i.e., type 1 diabetes (T1DM) and T2DM, showed a wide spectrum of effects with variable outcomes, indicating the presence of multiple factors that interact in shaping the outcome of CM intervention in patients with diabetes.

The glycemia-regulating and anti-diabetic prophylactic effects of CM were revealed early on, with zero prevalence of diabetes being reported in the habitually CM-consuming Raica community of northwest Rajasthan/India [22]. Agrawal and co-workers found that the prevalence of diabetes among the Raica community was negligible when compared with the non-consumers in the same Raica community. These values were in line with other non-Raica communities, where the prevalence of diabetes was 5.5% and 0.4% among non-consumers and consumers, respectively [22]. Since that time, a growing wealth of evidence at different levels of research, including in vitro and in vivo, along with clinical and observational studies, has been published, confirming the beneficial anti-diabetic or the anti-hyperglycemic ability of CM. Consistent with previous reports, in a systematic review on CM and diabetes, Mirmiran et al. [23] reported that most of the reviewed studies demonstrated favorable effects on DM, resulting in reduced insulin resistance and glucose levels. Nonetheless, the lack of quantitative assessment via meta-analysis and the relatively small number of included studies makes it difficult to ascertain such a positive effect. Therefore, the present meta-analysis was designed and implemented to obtain a more stable estimate of the effect size of CM intake on glucose homeostasis parameters in patients with diabetes, assess the generalizability of results claiming CM as an effective remedy for patients with diabetes, examine the variability between studies, and perform subgroup analyses for potential moderators, e.g., type of disease (T1DM or T2DM), type of CM (fresh or pasteurized/fermented), and duration of CM intake (>6 months or ≤6 months). Based on previous literature showing an inverse relationship between CM intake and the incidence of diabetes, and the apparent positive effects of CM intake on diabetes in patients and animal models, we hypothesized that CM intake by patients with diabetes will improve their glycemic control and glucose homeostasis markers in comparison with those patients on standard, usual care or other ruminant milk.

## 2. Materials and Methods

This systematic review and meta-analysis was conducted and reported according to the Preferred Reporting Items for Systematic Reviews and Meta-Analyses (PRISMA) [24]. The protocol of this study was registered with the International Prospective Register of Systematic Reviews (PROSPERO, CRD42021276157).

### 2.1. Inclusion Criteria

Experimental studies investigating the effect of CM intake on glucose homeostasis parameters were eligible for inclusion if they met the following criteria: (1) randomized controlled trials (RCTs) conducted on patients with T2DM and T1DM, (2) RCTs included subjects >18 years old, (3) RCTs that provided sufficient data on the baseline and final measures of fasting blood glucose (FBG), postprandial blood glucose (PBG), glycosylated hemoglobin (HbA1c), fasting serum insulin (FI), insulin resistance (expressed in terms of Homeostatic Model Assessment for Insulin Resistance, HOMA-IR), insulin dose (ID), serum insulin antibody (IA), and C-peptide (CP) in both CM consumers and control groups of patients with diabetes. We included only publications in the English language.

### 2.2. Exclusion Criteria

Observational abstracts, reviews, longitudinal studies, unpublished papers, and non-English articles were excluded. Moreover, RCTs conducted exclusively on children, healthy participants, athletes, pregnant, lactating women, and animals were also excluded from the review. In addition to studies with no sufficient data reporting on the outcome measures of interest, studies reporting the presence of comorbidities with diabetes were also excluded.

### 2.3. Database Search

Electronic databases were systematically searched by two authors (N.K., L.M.) to find relevant RCTs examining the impact of CM intake on glucose homeostasis parameters in patients with DM. These databases included Google Scholar, PubMed/MEDLINE, EBSCO host, CINAHL, ScienceDirect, Cochrane, ProQuest Medical, Web of Science, and Scopus databases from the database inception (1950) until the end of November 2021, without restriction on publication date. In addition to the above databases, we searched the grey literature for additional eligible studies. The reference lists of the included articles and related reviews were also manually checked. Searching terms included “Camel milk” OR “*dromedary* camel milk” OR “Arabian *camel milk*” AND “diabetes” OR “diabetes mellitus” OR “type 1 diabetes” OR “T1DM” OR “type 2 diabetes” OR “T2DM” OR “juvenile diabetes” OR “adulthood diabetes” AND “insulin” OR “glycemic control” OR “glucose homeostasis” OR “glucose” OR “glycosylated/glycated hemoglobin” OR “HbA1c” OR “Fasting blood glucose” OR “FBG” OR Postprandial blood glucose” OR “PBG”. In addition, all the reference lists of the included articles and related reviews were manually checked to avoid missing any relevant studies. The complete search strategy is indicated in Table 1.

### 2.4. Main Outcomes and Measures

The main outcome was to report the effect size of CM intake to patients with diabetes on glucose homeostasis parameters; namely, fasting blood glucose (FBG), postprandial blood glucose (PBG), glycosylated hemoglobin (HbA1c), fasting serum insulin (FI) levels, insulin resistance (expressed in terms of HOMA-IR), insulin dose (ID), serum insulin antibody (IA), and C-peptide (CP). To standardize data extraction, the review team systematically collected and coded data for study characteristics (e.g., author names, year of publication, country, sample size, and participants’ characteristics such as age, sex, or proportion of males) and the main findings for glucose homeostasis for both intervention and control groups (with *p*-values).

### 2.5. Data Extraction

Two investigators (N.K., M.K.) individually screened the records and extracted the data, while the other two authors (R.K., L.M.) double-checked the extracted data. Any disagreements were resolved by the chief investigator (M.F.). A screening tool was developed for the data extraction from each study: first author’s name, publication year, region of the study, the sample size in each group, sex of the participants, mean age, study design, parameters measured, type of DM, type of CM, amount consumed per unit time, duration of intervention, and the mean and standard deviation (SD) of outcome measures for the intervention and control groups, a summary of the significances.

### 2.6. Quality Assessment

All the included studies were assessed using the Cochrane risk of bias (ROB) assessment tool. This tool aims to make the process of assessing bias more accurate and clearer through the examination of six domains of bias: performance bias, selection bias, attrition bias, detection bias, reporting bias, and other biases [25,26]. All the selected articles were scored by 2 authors (M.K., L.M.). Disagreement between the authors was resolved by a third assessor (N.H.).

### 2.7. Data Synthesis and Statistical Analysis

A meta-analysis random-effects model was used for all statistical tests [27]. In the random-effects model, it was assumed that there was a distribution of true effect sizes rather than one true effect size [27]. We estimated the mean of this distribution of true effect sizes. *τ*^2^ statistics were used to assess heterogeneity within studies, and *I*^2^ statistics were used to assess the heterogeneity between included studies [28]. To assure that our meta-analysis findings were not driven by a single study, leave-one-out sensitivity analysis was conducted by iteratively eliminating one study at a time. Computing *I*^2^ and *τ*^2^ statistics were particularly important to examine heterogeneity [27,28]. *I*^2^ of less than 30% represented low heterogeneity, 30–59% represented moderate heterogeneity, 60–90% represented substantial heterogeneity, and more than 90% represent considerable heterogeneity [28]. Graphical plots were used to aid the interpretation of the results visually [29]. Funnel-plot-based analysis was used to detect publication bias, and the nonparametric trim and fill method was used to confirm the findings [30]. Finally, subgroup analyses were performed to investigate differences in the effect of CM intake between the main effectors presented as categorical variables (T1DM or T2DM, fresh or pasteurized/fermented CM, CM intake for >6 months or ≤6 months). Subgroup analysis was conducted for those glucose homeostasis parameters with at least ten articles included.

The effect sizes of all intended outcomes were expressed as mean differences (MDs) and 95% confidence interval (CI). The effect sizes were pooled, exerting a random-effects model with RevMan software (Review Manager, version 5.3.5; The Nordic Cochrane Center, The Cochrane Collaboration, 2014). The mean net changes (mean values ± standard deviation, SD) of all the variables between the CM intervention group and control group at the baseline and the final stage of the study were calculated. In the present study, SD was computed using the Cochrane handbook’s proposed equations as follows:$$\mathrm{From}\;\mathrm{Standard}\;\mathrm{error}\;\mathrm{when}\;\mathrm{SD}\;\mathrm{was}\;\mathrm{not}\;\mathrm{given},\;SD=SE\times \sqrt{N}.$$

When SD change was not given:$$SD_{E,change}=\sqrt{SD_{E,baseline}^{2}+SD_{E,final}^{2}-\left(2\times Corr\times SD_{E,baseline}\times SD_{E,final}\right)}$$

When the combination of intervention groups was required:$$\sqrt{\frac{\left(N_{1}-1\right)SD_{1}^{2}+\left(N_{2}-1\right)SD_{2}^{2}+\frac{N_{1}N_{2}}{N_{1}+N_{2}}\left(M_{1}^{2}+M_{2}^{2}-2M_{1}M_{2}\right)}{N_{1}+N_{2}-1}}$$

*I*^2^ was used to assess heterogeneity between studies. The *I*^2^ indicates the percentage of the variability in effect estimates across studies due to heterogeneity rather than sampling error (*I*^2^ > 50%: substantial [31] heterogeneity). Any potential publication bias was identified using the funnel plot. A *p*-value < 0.05 was considered statistically significant. Sensitivity analysis was undertaken by excluding a single study at a time, to examine the robustness of the overall findings to investigate the effect of the results on the meta-analysis.

## 3. Results

### 3.1. Study Selection

In the primary search, 4054 articles were identified (Figure 1). Duplicate checking led to the elimination of 3729 articles, while 325 articles remained for initial screening based on the title and abstract. A total of 21 articles were then selected for full-text screening, out of which 7 articles were excluded due to insufficient data reporting of the outcome measures of interest. Fourteen articles [32,33,34,35,36,37,38,39,40,41,42,43,44,45] were therefore included in the quantitative meta- and sub-group analysis.

### 3.2. Characteristics of Included Studies

Table 2 summarizes the characteristics of the included studies. The sample size of the eligible studies ranged from 12 to 250 subjects (total of 663), with ages between 8 and 70 years (excluding studies exclusively conducted on young patients <18 years). One study was conducted exclusively on males [44] and the rest on both sexes. The percent of males was 59.4%. Duration of intervention ranged between 2 to 24 months. Intake dosage varied between 250 mL and 500 mL daily or twice a week. The type of CM intake as described in 14 studies: fresh [32,33,35,37,39,40,41,42,44,45], fermented [38] and pasteurized [36,43]. Studies were accomplished in India [32,33,35], Iran [36,37,38,41], Yemen [39], Sudan [40], Egypt [42], Saudi Arabia [43], Libya [44], and China [45], and were published between 2003 and 2021. In the current meta-analysis, the control group that received the usual care of diabetes was used as the comparators in all the included studies. Eight studies [32,33,35,39,40,41,42,45] out of the 14 solicited articles advised the patients to follow a strict diet, exercise, and insulin treatment one month before the initiation of the intervention.

### 3.3. Effect of CM Intake on Glycemic Control Parameters

The pooled results obtained using a random-effects model showed a statistically significant decrease in HbA1c levels (MD, −1.24, 95% CI: −2.00, −0.48, *p* < 0.001, *I*^2^ = 94%) (Figure 2; Appendix A) and ID received (MD, −16.72, 95% CI: −22.09, −11.35, *p* < 0.00001, *I*^2^ = 90%) (Figure 3; Appendix A) in patients with diabetes supplemented with CM in comparison to those receiving the usual care of diabetes. A clear tendency, but without significance, was observed in those patients supplemented with CM to have decreased FBG (MD, −23.32, 95% CI: −47.33, 0.70, *p* = 0.06, *I*^2^ = 98%) in comparison with those receiving usual care (Figure 4; Appendix A). Conversely, patients supplemented with CM did not show statistically significant reductions in PBG (MD, −34.14, 95% CI: −75.26, 6.98, *p* = 0.1, *I*^2^ = 99%) (Figure 5; Appendix A), FI levels (MD, −0.40, 95% CI: −3.33, 2.52, *p* = 0.79, *I*^2^ = 97%) (Figure 6; Appendix A), HOMA-IR (MD, −0.69, 95% CI: −2.58, 1.20, *p* = 0.47, *I*^2^ = 90%) (Figure 7; Appendix A), IA (MD, −1.13, 95% CI: −4.73, 2.47, *p* = 0.54, *I*^2^ = 89%) (Figure 8; Appendix A) and CP (MD, 0.01, 95% CI: −0.11, 0.14, *p* = 0.82, *I*^2^ = 93%) (Figure 9; Appendix A) when compared with those patients receiving usual care.

### 3.4. Subgroup Analysis

Subgroup analysis was performed for those parameters with at least ten included articles. We stratified studies based on the type of diabetes (T1DM or T2DM), type of CM used (fresh or pasteurized/fermented), and duration of the intervention (>6 months or ≤6 months). It was intended that subgroup analysis for the glucose homeostasis parameters significantly affected by CM intervention; namely HbA1c and insulin dose (ID), be carried out. However, the lack of a sufficient number of studies for ID rendered this analysis unattainable. Instead, and due to the availability of a required number of ten studies for both FBG and HbA1c, with the reported clear tendency of lower FBG by CM, subgroup analysis was performed only for the latter parameters (FBG and HbA1c). For FBG (mg/dL), the results of subgroup analysis unraveled that T2DM is mostly affected by CM intervention (MD, −15.62, 95% CI: −26.71, −4.54, *p*-value = 0.006, *I*^2^ = 66%) (Figure 10). Nonetheless, intervention duration and type of CM did not show significant effects on FBG (Figure 11 and Figure 12, respectively) (Table 3).

Patients with T1DM received more benefits from consuming CM than T2DM in lowering the HbA1c (MD, −1.21, 95% CI: −2.24, −0.19, *p*-value = 0.02, *I*^2^ = 92%) (Figure 13), with clear tendency, but not significant, for patients with T2DM. Both treated (Pasteurized/Fermented) and fresh CM gave similar beneficial effects in lowering HbA1c (−0.31, 95% CI: −0.45, −0.18, *p*-value = 0.00001, *I*^2^ = 0.0% and −1.50, 95% CI: −2.26, −0.74, *p*-value = 0.00001, *I*^2^ = 85%, respectively) (Figure 14). Lastly, both short and long-term intervention durations (≤6 months and >6 months) gave significant reductions in HbA1c (MD, −1.21, 95% CI: −2.18, −0.23, *p* = 0.02, *I*^2^ = 95%, %, and MD, −1.36, 95% CI: −2.19, −0.53], *p* = 0.001, *I*^2^ = 71%, respectively) with relatively superior effect for longer duration than shorter duration (Figure 15) (Table 3).

### 3.5. Quality Assessment and Publication Bias

Figure 16a,b shows the risk of bias graph and summary, respectively. Three studies performed adequate sequence generation (Agrawal et al., 2011a; Ejtahad et al., 2015; Fallah et al., 2018). Participants’ allocation was adequately concealed in one study (Fallah et al., 2018). Blinding of participants and key study personnel was ensured in two studies (Abdalla and Fadlalla, 2018; Fallah et al., 2018). Outcome assessors were not blinded in all the included studies, except in Ejtahad et al., (2015) and Fallah et al., (2018). Incomplete data outcome was adequately addressed in two studies (Ejtahad et al., 2015; Mohamad et al., 2009). In all the included studies, the expected outcomes were reported as pre-specified in the methodology, except in Mostafa et al., 2014, where no sufficient information permitted the judgment of the latter. More than half of the included studies (8/14, 57%) had a potential source of bias related to the study design and/or protocol (Agrawal et al., 2003; Agrawal et al., 2005 Agrawal et al., 2011b; Margdarinejad et al., 2021; Mohamad et al., 2009; Mostafa and Al- Musa, 2014; Shareha et al., 2016; Wang et al., 2009).

### 3.6. Sensitivity Analysis

When a sensitivity analysis was performed for articles on FBG, the removal of Abdalla and Fadlalla’s study [40] did not change the heterogeneity, however, it did affect the overall effect (−26.19, 95% CI: −50.66, −1.71, *p* = 0.04). Likewise, the elimination of Mohamad et al., 2009 study [42] led to a statistically significant overall effect (−13.64, 95% CI: −21.45, −5.84, *p* < 0.001, *I*^2^ = 77%). Consistent with the findings of FBG levels, when a sensitivity analysis was performed for articles on PBG, the elimination of Abdalla and Fadlalla’s study [40] led to a significant overall effect with no change in heterogeneity (- 45.26, 95% CI: −88.49, −2.03, *p* = 0.04). For HbA1c, FI, HOMA-IR, and IA, when a sensitivity analysis was performed, no changes were found in the results. For ID, when a sensitivity analysis was performed, the elimination of Abdalla and Fadlalla [40] led to a low heterogeneity of 26% with no impact on the overall effect. For the C-peptide, no changes in the results were detected after a sensitivity analysis was carried out.

## 4. Discussion

According to the available literature and the best of our knowledge, this is the first systematic review and meta-analysis examining the effect size of CM intake on glucose homeostasis parameters in patients with diabetes. Our meta-analysis revealed that the supplementary intake of CM induced significant effects on HbA1c and ID, with insignificant enhancements in FBG, PBG, FI, HOMA-IR, CP, and IA in patients with T1DM and T2DM when compared to usual diabetic care. Moreover, subgroup analyses showed that patients with T2DM receive more benefits than those with T1DM in lowering their FBG by CM intake, whereas intervention duration and type of CM did not show significant effects on FBG levels. Meanwhile, patients with T1DM receive more benefits from consuming CM than those with T2DM in lowering the HbA1c, while both fresh and treated (pasteurized/fermented) CM gave similar beneficial effects in lowering HbA1c. Lastly, both short and long-term intervention durations (>6 months or ≤6 months) gave significant reductions in HbA1c, with a relatively superior effect for a longer duration.

The most two prominent effects of CM intake in patients with diabetes in our current work are the reduction of HbA1c and ID, along with a clear, but not significant, lowering effect on FBG. These evident effects are a mirror for the previous notion that CM could play a role in lowering the risk of diabetes, decreasing its prevalence, and being an adjuvant therapy for diagnosed patients. Furthermore, these findings are in line with previous findings of laboratory animal studies including dogs [46,47] and rodents [48,49,50,51,52,53,54], which revealed reductions in blood glucose and HbA1c, as well as C-peptide and other diabetes-related parameters. Most recently, the antidiabetic effect of CM in a diabetes mouse model, by simultaneous measurement of blood glycemia, showed that blood glucose and HbA1c were significantly reduced compared to that in the diabetic control group. Interestingly, researchers found that the therapeutic effect of CM was completely comparable with the glibenclamide antidiabetic drug, suggesting that CM could be used as an alternative regimen in the medical nutrition therapy of diabetes [55].

It was speculated that the anti-diabetic properties of CM are due to the proteins in CM. The antidiabetic effect of CM protein was extensively reviewed by Malik and co-authors, who reported various properties of CM in ameliorating hyperglycemia in patients with diabetes. The potential mechanisms could be summarized as follows: (i) the presence of insulin in CM possesses special properties which make its absorption into blood circulation easier than insulin from other sources, or make it resistant to proteolysis; (ii) CM insulin is encapsulated in nanoparticles (lipid vesicles) that make its passage possible through the stomach and entry into the circulation; (iii) some other elements of CM induce antidiabetic properties. The sequence of CM insulin and its predicted digestion pattern does not suggest differentiability to overcome the mucosal barriers before being degraded and reaching the bloodstream. However, researchers cannot exclude the possibility that insulin in CM is present in nanoparticles capable of transporting this hormone into the bloodstream. Another more probable explanation is that CM contains small “insulin-like” molecule substances that mimic insulin interaction with its receptor [56].

Several reviews tried to synopsize the available evidence pertaining to the molecular and cellular mechanisms underlying the anti-diabetic, anti-hyperglycemic effect of CM [15,16,23,56,57,58,59,60]. In addition, different hypotheses and suggested mechanisms have been evoked to explain the cumulative notions on the anti-hyperglycemic, glycemic-normalizing, and diabetes ameliorating effects of CM. Among these, persistent results have shown that CM possesses insulin-like hormonal activity, which decreases the requirement of exogenous insulin in patients with T1DM [61,62]. Such a notion helps in explaining our finding on the significant reduction in ID in patients with diabetes taking insulin injections. This is further supported by the clinical research on the consumption of CM in patients with T1DM, which indicated that consuming CM daily decreased FBG and reduced the mean ID required by 37% (from 30.40 ± 11.97 to 19.12 ± 13.39 u/day) [63]. A possible explanation is that a considerable quantity of insulin, reaching 52 units/liter is detected in CM, using the radioimmunoassay [35]. However, a recent insulin immunoreactivity analysis for CM samples revealed the lack of a high quantity of insulin (below detection range with the anti-human insulin antibody), leading the authors to conclude that the hypoglycemic effect of CM might be ascribed to other components rather than insulin [64]. Furthermore, the multiplex panel assay showed that CM samples possessed insulinotropic polypeptide (gastric inhibitory polypeptide, GIP) and have higher immunoreactivity to visfatin, resistin, and ghrelin than other tested ruminant milk samples [64].

Other proteins in higher concentrations in CM than in other milk could interact with the insulin receptor and contribute by their antioxidant and anti-inflammatory effect to the regeneration of β-cells in the pancreas [21]. The small particles of insulin-like molecules in CM have similar interactions with insulin receptors [56]. In addition, the special features of CM protein (lysozyme, lactoferrin, and lactoperoxidase) are indigestible by the stomach enzyme (pepsin); hence, those proteins are not coagulated at a low pH.

The anti-hyperglycemic effect of CM may involve complex molecular and cellular mechanisms that affect insulin synthesis and secretion, as well as glucose metabolism and transport. Among these, the two most important factors that may shape the way CM affects diabetes are insulin receptor function and insulin synthesis and secretion by the pancreatic β-cells, glucose transport in the insulin-sensitive tissues, and, lastly, survival, growth, and the overall activity of the pancreatic cells [57]. Furthermore, Abdulrahman and co-workers [65] found that the peptide/protein nature of the active component in CM demonstrates an allosteric effect on the insulin receptor, with differential effects on its intracellular signaling pathways. In more recent work, the CM-ameliorating effect on serum glycemic parameters in chemically induced diabetic rats was found to be associated with up-and-down-regulation in the mRNA expressions of a set of genes involved in glycemic control. Interestingly, CM intake exhibited a superior effect over metformin in downregulating *CPT-1* gene expression, a gene that is upregulated in diabetic rats. Finally, CM administration improved pancreatic β-cell functioning, as revealed by restoring the immunostaining reactivity of GLUT-4 and insulin in the pancreas of diabetic rats [66], a finding that is consistent with a previous publication by Agrawal and co-workers on pancreatic β-cell of the pancreas [63]. At the genetic level, CM normalizes the histopathology of patients with diabetes, affects vital enzymes and proteins for cardiovascular and hepatorenal functions, controls phosphoenolpyruvate carboxykinase (*PEPCK*) gene transfer that induces a gluconeogenesis effect, and plays a pivotal role in regulating a set of metabolizing enzymes that affect carbohydrates and lipids metabolism [58].

Being the “ships of the deserts”, camels depend on eating domestic desert plants which include distinctive phytochemicals such as phenolic compounds (e.g., flavonoids, phenolic acids, tannins, and quinones). These phytochemicals are funneled in their milk, and in turn, might have anti-diabetic properties [67,68]. Another aspect of the functional properties of CM against hyperglycemia is ascribed to its considerable quantities of zinc, which gives CM superiority over other ruminant milk in controlling the secretory activity of islets of β pancreatic cells islets and in insulin biosynthesis secretion saturation [66,69]. Finally, the ability of CM to ameliorate the levels of elevated hormones, TNF-α and TGF-β1, produced in response to diabetes, is one of the underlying mechanisms for such a protective effect. All these benefits behind CM intake allow it to play a role in mitigating the risk of diabetic complications among patients with diabetes [58] and the side effects for the long-term excessive use of insulin injections [70].

Recently, a list of bioactive peptides with antidiabetic potential has been identified in dietary proteins, which could improve insulin uptake, decrease blood glucose levels, and inhibit key enzymes involved in the development and progression of diabetes [11,71]. Among these, peptides produced from trypsin-digested CM proteins have been shown to inhibit dipeptidyl peptidase IV (DPPIV, DPP-4) or T-cell antigen CD26, a key enzyme regulating the biological activity of the incretin hormone, glucagon-like peptide-1, which plays an important role in glucose homeostasis [72,73,74,75,76]. Furthermore, short peptides produced from the alcalase, papain, and bromelain-induced proteolysis showed in vitro inhibited effects against pancreatic α-amylase [75].

In a comprehensive review on the in vivo evidence pertaining to the anti-hyperglycemic potential of CM [59], authors reported that CM immunoglobulins with small weight and size may offer a wide potential through interaction with the host cell protein and induction of regulatory cells, which results in downregulating the immune system, and ends with β-cell salvage [63,77]. Some works suggest that the insulin-like proteins in CM can resist proteolysis, thus facilitating its absorption into the bloodstream faster than the insulin-like proteins from other milk sources. This could be explained in terms of CM protein’s resistance against coagulation in the acidic environment of the stomach, and its higher buffering capacity than that of other ruminants’ milk [51]. In addition, the anti-diabetic or glycemia-controlling effect of CM is partially ascribed to the well-known α-amylase and α-glucosidase inhibitory effects [78,79]. Current research highlighted that the ability of CM to inhibit α-amylase and α-glucosidase alleviates carbohydrate digestion and hydrolysis, leading to reduced sugar absorption by the human intestines [80].

Fermentation of CM has been found to enhance its functional properties, i.e., antioxidant, anti-hypertensive, anti-proliferative ad antidiabetic effects [78,81,82]. Fermentation of CM was found to increase the α-amylase and α-glucosidase inhibitory potential, the two enzymes that are known to be involved in carbohydrates digestion; their inhibition can affectively mitigate the elevated blood sugar among patients with diabetes via diminishing carbohydrate hydrolysis [83]. For example, probiotic lactic acid bacteria isolated from CM can exhibit remarkable probiotic and exopolysaccharide-producing characteristics, which in turn aid in enhancing the antidiabetic effect of the fermented CM [84]. It is commonly practiced to drink CM in its fresh, un-heat-treated form, or fermented sour form. On the other hand, heat treatments such as boiling, pasteurization, or sterilization are among the most effective thermal preservation processes that aid in preserving the integrity and conserving the safety of the ruminant milk including CM. These heat treatments are essential to prevent spoilage of CM and the avoidance of being a carrier for food-borne illnesses and food intoxication. Although CM protein and other bioactive chemicals are sensitive to high temperature, particularly after boiling at high temperature; CM α- immunoglobulin and lactalbumin are more heat tolerant, conversely to lactoferrin from other ruminants milk, such as bovine milk, which gets denatured [46].

Thermal and un-thermal treatments have been found to directly influence the biological, microbiological, nutritional, and functional properties of CM and its proteins [9,14,38,78,81,82,85,86] in a way that is expected to differently affect the antidiabetic potential CM. However, the current work failed to show a significant effect for the treated (pasteurized/fermented) CM against the elevated glucose homeostasis markers characterized by the patients with diabetes. This could be ascribed to the lack of sufficient studies on these treatments and their effects on CM and glucose homeostasis, and the methodological approach we followed in combining the two types of treatments (heat and non-heat or fermentation treatments) in the subgroup analysis.

The current work entailed several strengths as being the first meta-analysis in this domain, having assessed several glucose homeostasis parameters, with the stratification of the analysis by type of diabetes, type of CM, and duration of intervention. However, the current work entails several limitations that should be considered when interpreting the current findings. First, there was high methodological and statistical heterogeneity between the selected articles. The latter could be ascribed to various factors, including but not limited to the differences in the study designs, intervention durations, type and amount of CM administered, the type of diabetes, medications used along with the administered CM, age of the study subjects, their sex, and time since diagnosis with diabetes. This dictates the need for future controlled trials with consistent and fixed study elements to accurately elucidate the effect of CM on glucose homeostasis and minimize the effect of variable confounders and interfering factors. Considering the evident mechanisms underpinning the glucose-lowering and the insulin-like effect of CM may prompt clinicians to consider the daily use of two cups of pasteurized CM in patients with T1DM and T2DM as a safe, efficient and effective adjuvant therapy. It is inferred that such adjuvant therapy may reduce the treatment costs and further reduce the dosage of glucose-lowering medications and insulin injections, resulting in less plausible adverse effects.

## 5. Conclusions

To conclude, CM could be used as effective adjuvant therapy for patients with both types of diabetes, effectively reducing the short-term and long-term hyperglycemia parameters, i.e., fasting blood glucose and HbA1c, respectively. Owing to the bioactive peptide and hormone-like proteins involved in CM, the insulin dose required for patients with diabetes could be reduced by the regular long-duration administration of CM. Long-term, more controlled clinical trials are warranted to overcome the raised limitations presented in the high heterogeneity of the analyzed articles and to provide evidence of a more robust conclusive effect on the impact of CM intake of patients with diabetes.

## Figures and Tables

**Figure 1 nutrients-14-01245-f001:**
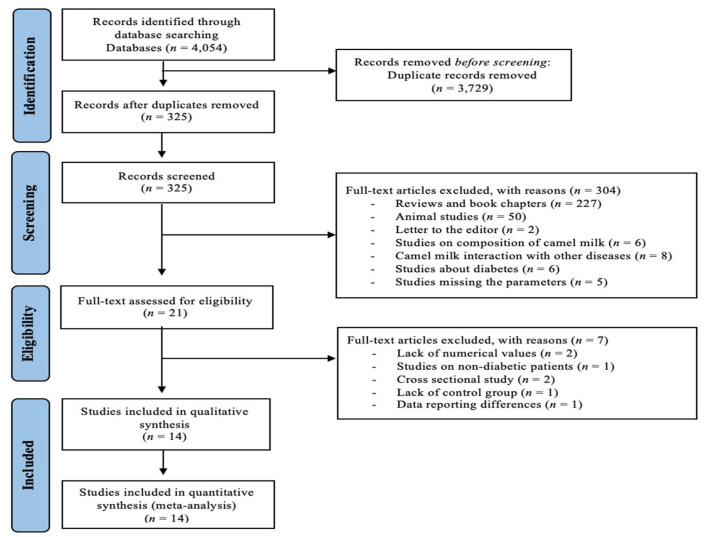
PRISMA flow diagram for study selection on the effect of CM on glucose homeostasis parameters.

**Figure 2 nutrients-14-01245-f002:**
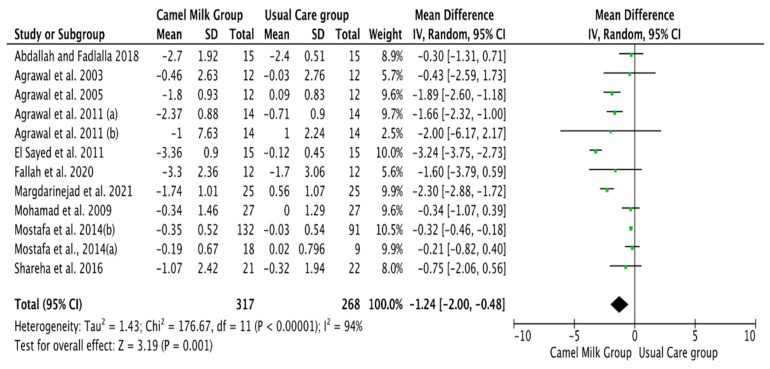
Forest plot for the effect of CM intake on glycosylated hemoglobin (HbA1c). Note: Mostafa and Al-Musa, 2014 (a) for T1DM and (b) for T2DM.

**Figure 3 nutrients-14-01245-f003:**
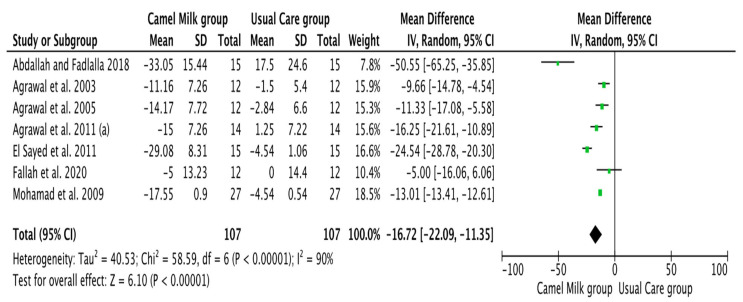
Forest plot for the effect of CM intake on insulin dose (ID). (a): difference between the two articles that are published by same author (Agrawal), same year.

**Figure 4 nutrients-14-01245-f004:**
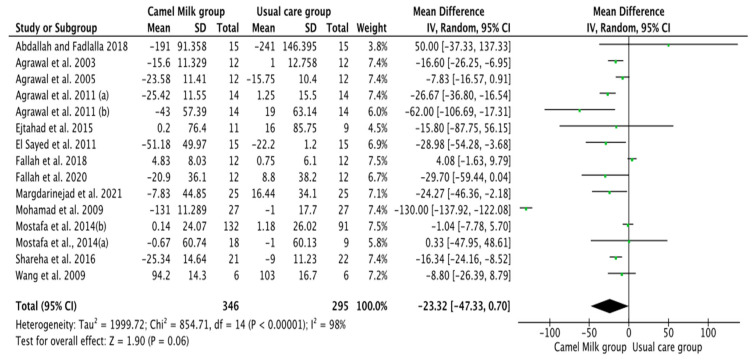
Forest plot for the effect of CM intake on fasting blood glucose (FBG). Note: Mostafa and Al-Musa, 2014 (a) for T1DM and (b) for T2DM.

**Figure 5 nutrients-14-01245-f005:**
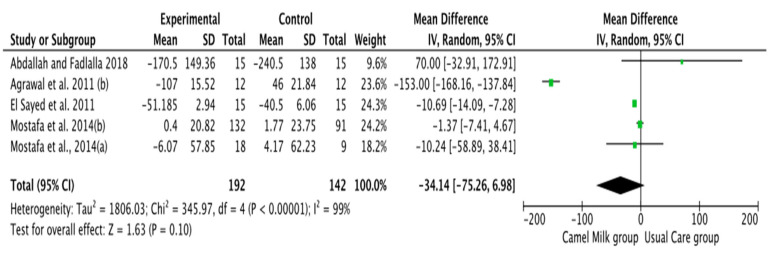
Forest plot for the effect of CM intake on postprandial blood glucose (PBG). Note: Mostafa and Al-Musa, 2014 (a) for T1DM and (b) for T2DM.

**Figure 6 nutrients-14-01245-f006:**
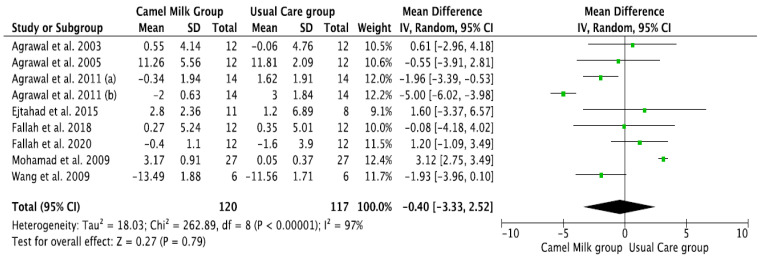
Forest plot for the effect of CM intake on fasting serum insulin (FI). (a) and (b) are two different studies (difference between the two articles that are published by same author (Agrawal), same year).

**Figure 7 nutrients-14-01245-f007:**
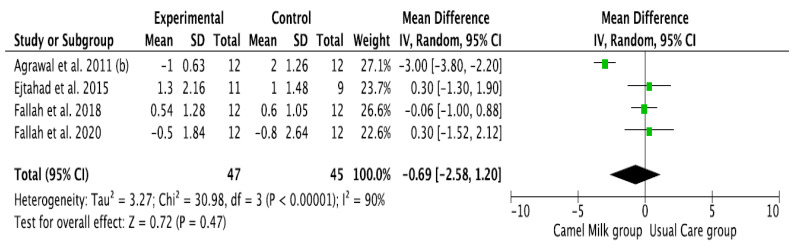
Forest plot for the effect of CM intake on HOMA-IR. (b): difference between the two articles that are published by same author (Agrawal), same year.

**Figure 8 nutrients-14-01245-f008:**
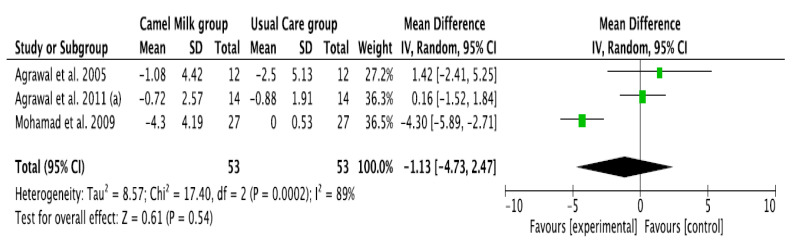
Forest plot for the effect of CM intake on circulating insulin antibody (IA). (a): difference between the two articles that are published by same author (Agrawal), same year.

**Figure 9 nutrients-14-01245-f009:**
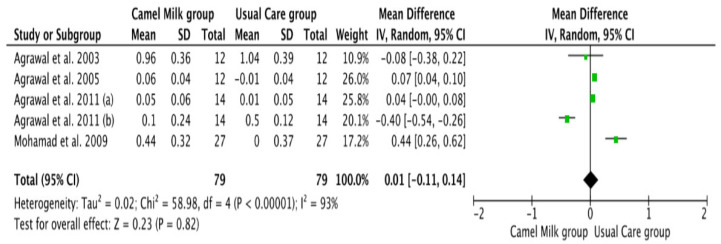
Forest plot for the effect of CM intake on C-peptide (CP). (a) and (b) are two different studies (difference between the two articles that are published by same author (Agrawal), same year).

**Figure 10 nutrients-14-01245-f010:**
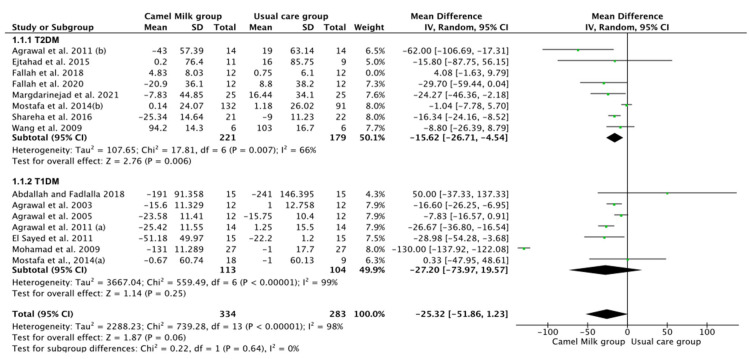
Forest plot for the effect of CM intake on FBG levels based on the type of diabetes. Note: Mostafa and Al-Musa, 2014 (a) for T1DM and (b) for T2DM.

**Figure 11 nutrients-14-01245-f011:**
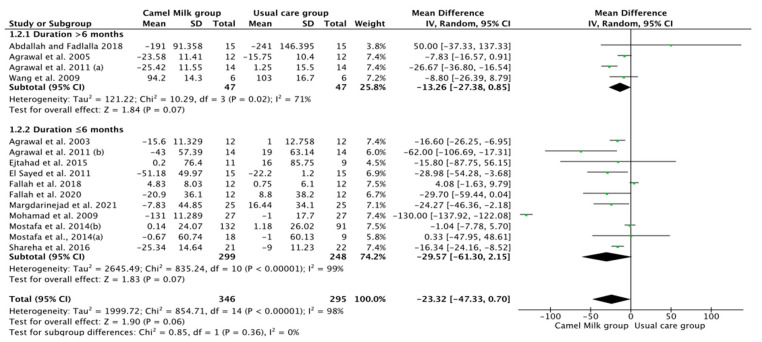
Forest plot for the effect of CM intake on FBG based on intervention duration. Note: Mostafa and Al-Musa, 2014 (a) for T1DM and (b) for T2DM.

**Figure 12 nutrients-14-01245-f012:**
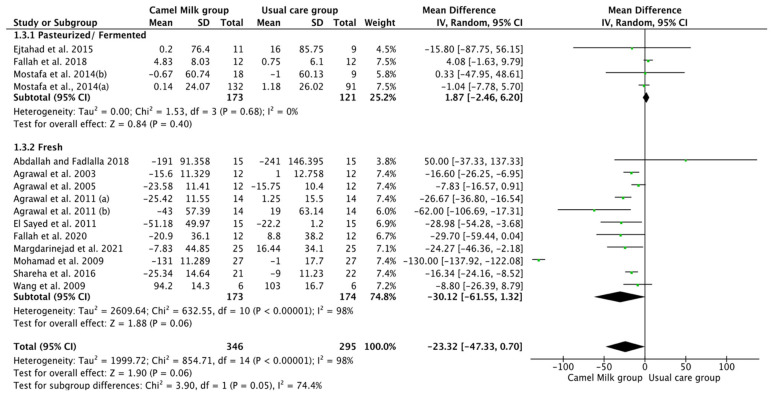
Forest plot for the effect of CM intake on FBG levels based on the type of CM. Note: Mostafa and Al-Musa, 2014 (a) for T1DM and (b) for T2DM.

**Figure 13 nutrients-14-01245-f013:**
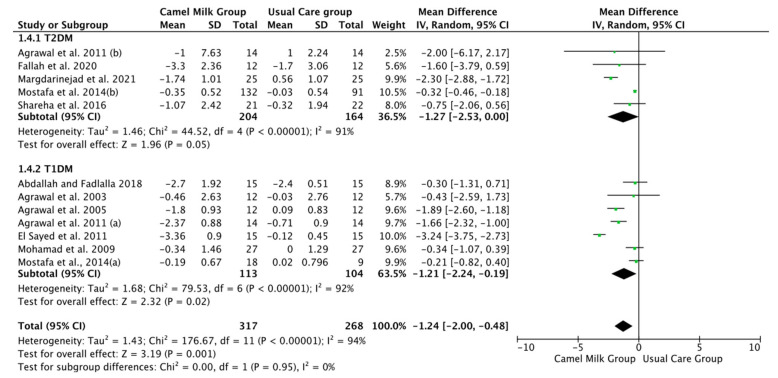
Forest plot for the effect of CM intake on HbA1c levels based on the type of diabetes. Note: Mostafa and Al-Musa, 2014 (a) for T1DM and (b) for T2DM.

**Figure 14 nutrients-14-01245-f014:**
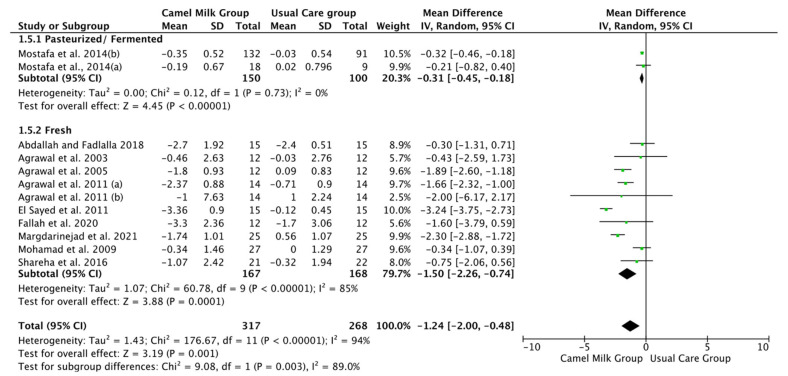
Forest plot for the effect of CM intake on HbA1C levels based on the type of CM. Note: Mostafa and Al-Musa, 2014 (a) for T1DM and (b) for T2DM.

**Figure 15 nutrients-14-01245-f015:**
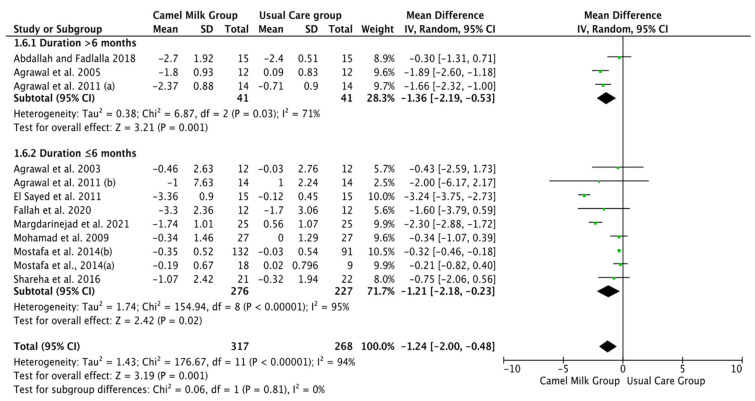
Forest plot for the effect of CM intake on HbA1c levels based on the intervention duration. Note: Mostafa and Al-Musa, 2014 (a) for T1DM and (b) for T2DM.

**Figure 16 nutrients-14-01245-f016:**
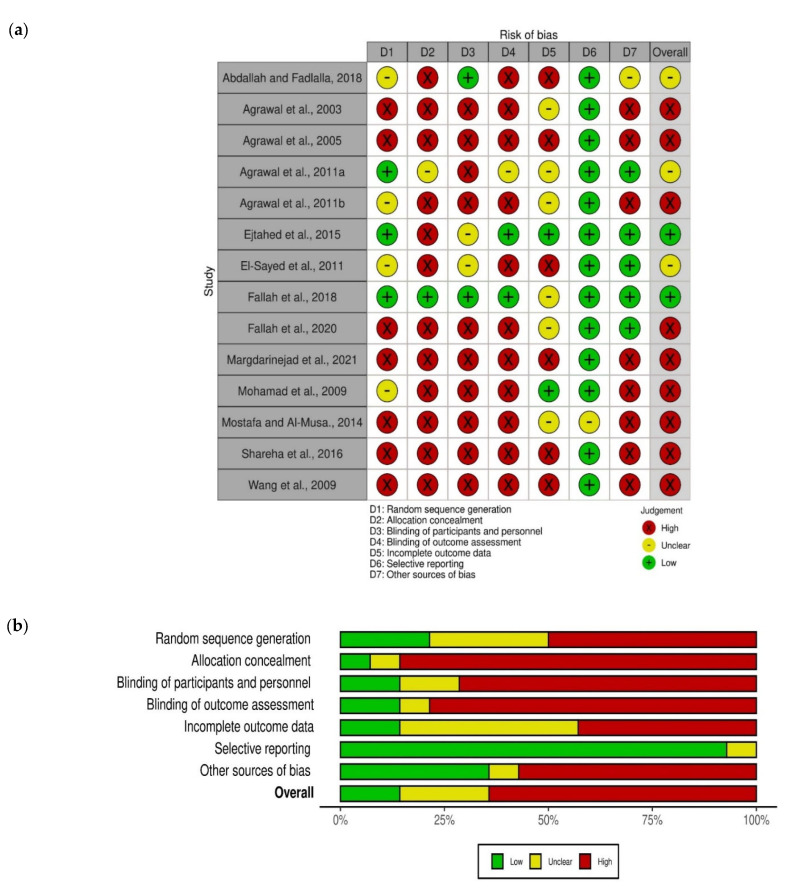
Risk of bias (**a**) Risk of bias summary: each risk of bias item for each included study; (**b**) risk of bias graph: each risk of bias item presented as percentages across all included studies. Green: Low risk of bias; Yellow Unclear risk of bias; Red High risk of bias (*n* = 14).

**Table 1 nutrients-14-01245-t001:** A summary of the search strategy adopted in the present systematic review and meta-analysis assessing the effects of camel milk (CM) intake on glycemic control among patients with diabetes.

Search Strategy Item	Search Strategy Details
String of keywords	“Camel milk” OR “*dromedary* camel milk” OR “Arabian *camel milk*” AND “diabetes” OR “diabetes mellitus” OR “type 1 diabetes” OR “T1DM” OR “type 2 diabetes” OR “T2DM” OR “juvenile diabetes” OR “adulthood diabetes” AND “insulin” OR “glycemic control” OR “glucose homeostasis” OR “glucose” OR “glycosylated/glycated hemoglobin” OR “HbA1c” OR “Fasting blood glucose” OR “FBG” OR Postprandial blood glucose” OR “PBG”.
Searched databases	Google Scholar, PubMed/MEDLINE, EBSCOhost, CINAHL, ScienceDirect, Cochrane, ProQuest Medical, Web of Science, and Scopus databases
Inclusion criteria	P (People): All patients with diabetes (T1DM, T2DM), including males/females, >18 years age group, from unspecified ethnic/racial backgrounds. I (Intervention/exposure): Intake of camel milk (CM), in any form (fresh, dried/reconstituted, fermented/cultured) for any time duration. C (Comparison): Comparing consumers with non-consumers of CM, routine or usual diabetes care. O (Outcome): Effect size of consuming CM on glycemic control in patients with diabetes, fasting blood glucose (FBG), postprandial blood glucose (PBG), glycosylated hemoglobin (HbA1c), fasting serum insulin (FI), insulin resistance (expressed in terms of HOMA-IR), insulin dose (ID), serum insulin antibody (IA), and C-peptide (CP) S (study type): Original research, experimental/randomized controlled trial (RCT) study is eligible for inclusion.
Exclusion criteria	P (People): Healthy, non-diabetic people, studies exclusively on children with diabetes, athletes, pregnant, lactating, patients with other comorbidities I (Intervention/exposure): Non-CM. C (Comparison): Non-diabetes comparator. O (Outcome): Outcomes not described in sufficient numerical details for the glycemic control measures (using curves, graphs without numerical presentations). S (study type): Editorials, paper abstracts, case reports, commentaries, expert opinions, letters to the editor, reviews, conference abstracts or proceedings; non-peer-reviewed and unpublished data.
Moderators for meta-regression	Continuous, including the age of patients, time duration of CM intake (days/weeks/months) Dichotomous, including sex (male/female) and type of diabetes (T1DM, T2DM), duration of intake (more than 6 months, equal or less than 6 months).
Time filter	None applied (search from inception)
Language filter	English language only
Hand-searched target journals	Foods, Nutrients, International Dairy Journal, BMC Complementary Medicine and Therapies

**Table 2 nutrients-14-01245-t002:** Characteristics and major findings of the included studies on the effect of camel milk (CM) intake on glucose homeostasis parameters in patients with diabetes.

Authors, Publication Year	Country/City	Sample Size n (%Male)	Mean Age/Age Range (Year)	Study Design	Tested Glucose Homeostasis Parameters	Type of Diabetes (Type 1; Type 2)	Type of CM (Fresh or Pasteurized/Fermented)	Quantity of CM Consumed (mL/day) by CM Group	Duration of Intervention (>6 Months, ≤6 Months)	Parameters of the CM Group		Parameters of the Control Group		Results (CM Group Consumption Compared with Control)
										Before Treatment	After Treatment	Before Treatment	After Treatment	
Margdarinejad et al., 2021 [41]	Iran (Gorgan)	49 (44.9)	Age: >18	Randomized, case-control clinical trial	FBG, HbA1c	T2DM	Fresh	500	60 days (<6 months)	FBG: 163.08 ± 73.81 HbA1c: 7.56 ± 1.60	FBG: 155.25 ± 51.95 HbA1c: (5.82 ± 0.94	FBG: 135.84 ± 54.49 HbA1c: 7.48 ± 1.61	FBG: 152.28 ± 53.31 HbA1c: 8.06 ± 1.79	- CM: HbA1c significantly decreased; FBG insignificantly decreased - Control group: HbA1c significantly increased; FBG insignificantly increased.
Fallah et al., 2020 [37]	Iran (Tehran)	36 (36.11)	Age range: 30–70	Randomized parallel-group clinical trial	FBG, FI, ID, HbA1c, HOMA-IR	T2DM	Fresh	500	90 days (<6 months)	FBG: 169.3 ± 78.9 FI: 3.8 ± 2.8 ID: 31 ± 22 HbA1c: 12.7 ± 2.6 HOMA-IR: 1.7 ± 2	FBG: 148.4 ± 59.5 FI: 3.3 ± 0.4 ID: 26 ± 16.7 HbA1c: 9.4 ± 0.3 HOMA-IR: 1.2 ± 0.2	FBG: 143.2 ± 56.5 FI: 4.5 ± 4.1 ID: 43 ± 22.4 HbA1c: 11.2 ± 3.3 HOMA-IR: 2 ± 2.8	FBG: 152 ± 51.4 FI: 2.9 ± 0.4 ID: 43 ± 23.1 HbA1c: 9.5 ± 0.3 HOMA-IR: 1.2 ± 0.2	- FBG: reduced in CM; in the control group: no significant change. - A significant change in HbA1c in CM and control. - Changes of HbA1c, insulin, and HOMA-IR: insignificant between the two groups. - CM: ID reduced significantly.
Fallah et al., 2018 [38]	Iran	24 (42)	Age Range: 11–18; Mean: 13.77	Randomized, double-blind, crossover, controlled clinical trial	FBG, FI, HOMA-IR	Pre-diabetes	Fermented	250	112 days (<6 months)	FBG: 89.83 ± 7.14 FI: 2.82 ± 1.31 HOMA-IR: 3.78 ± 1.85	FBG: 94.66 ± 8.03 FI: 3.12 ± 0.66 HOMA-IR: 4.32 ± 0.83	FBG: 89.21 ± 8.64 FI: 2.57 ± 1.34 HOMA-IR: 3.34 ± 1.69	FBG: 89.96 ± 6.10 FI: 2.97 ± 1.30 HOMA-IR: 3.94 ± 1.66	- FI: showed an insignificant increase. - FBG and HOMA-IR changes: insignificant.
Abdalla and Fadlalla, 2018 [40]	Sudan	30 (26.67)	Age range: 8–19	Randomized, open case-control, parallel	FBG, PBG, ID, HbA1c	T1DM	Fresh	500	365 days (>6 months)	FBG: 286 ± 108 PBG: 264 ± 136 ID: 75.8 ± 25.5 HbA1c: 7.3 ± 2.9	FBG: 95 ± 22 PBG: 93.5 ± 17.5 ID: 42.75 ± 22.5 HbA1c: 4.6 ± 1.5	FBG: 335.5 ± 158.5 PBG: 334.5 ± 149.5 ID: 56.5 ± 32.5 HbA1c: 8.15 ± 0.85	FBG: 94.5 ± 15.5 PBG: 93.5 ± 14.5 ID: 74 ± 41 HbA1c: 5.75 ± 0.75	- CM: a significant reduction in ID by 46%; FBG: reduced by 67%; PBG: reduced by 65%; HbA1c: reduced by 37%. - Control group: glucose parameters were unchanged; IDs: increased after the 4 weeks.
Shareha et al., 2016 [44]	Libya (Tripoli)	43 (100)	Age range: 40–65	Randomized as study	FBG, HbA1c	T2DM	Fresh	500	90 days (<6 months)	FBG: 193.86 ± 5.29 HbA1c: 8.11 ± 0.50	FBG: 168.52 ± 3.88 HbA1c: 7.04 ± 0.07	FBG: 202.18 ± 3.67 HbA1c: 8.05 ± 0.57	FBG: 193.18 ± 3.12 HbA1c: 7.73 ± 0.10	- CM: FBG and HbA1c decreased significantly. - FBG reduced by 13.07%, - Control FBG reduced by 4.45%.
Ejtahed et al., 2015 [36]	Iran (Tehran)	20 (30)	Age range: 20–70	Randomized single-blinded controlled clinical trial	FBG, FI, HOMA-IR	T2DM	Pasteurized	500	60 days (<6 months)	FBG: 168.84 ± 50.94 FI: 10.77 ± 13.20 HOMA-IR: 3.4 ± 2.9	FBG: 169.92 ± 45.90 FI: 14.01 ± 13.31 HOMA-IR: 4.7 ± 3.6	FBG: 145 ± 43 FI: 10.30 ± 6.60 HOMA-IR: 3.0 ± 2.4	FBG: 161 ± 58 FI: 11.69 ± 6.25 HOMA-IR: 4.0 ± 2.3	- In CM group: increase of HOMA-IR - No changes in FBG and FI in both groups. - A significant increase in Insulin concentration in the CM group compared with the control. - An increase in HOMA-IR in both groups but no significant difference between the two groups.
Mostafa and Al-Musa, 2014 [43]	Saudi Arabia (Abha City)	250 (52.80)	Mean age: 38	Randomized, non-blinded, control trial	FBG, PBG, HbA1c	T1 and T2 DM	Pasteurized	250 mL (twice a week)	183 days (≤6 months)	T1DM: FBG: 211.67 ± 94.68 PBG: 198.61 ± 91.48 HbA1c: 8.55 ± 1.11 T2DM: FBG: 152.36 ± 38.50 PBG: 130.50 ± 32.37 HbA1c: 8.16 ± 0.82	T1DM: FBG: 211.00 ± 94.36 PBG: 192.54 ± 91.48 HbA1c: 8.36 ± 1.01 T2DM: FBG: 152.50 ± 37.59 PBG: 132.90 ± 33.44 HbA1c: 7.81 ± 0.84	T1DM: FBG: 209 ± 97.27 PBG: 207.22 ± 99.01 HbA1c: 8.99 ± 1.26 T2DM: FBG: 151.77 ± 41.35 PBG: 151.30 ± 36.77 HbA1c: 8.15 ± 0.86	T1DM: FBG: 208.61 ± 96.04 PBG: 211.39 ± 97.76 HbA1c: 9.01 ± 1.26 T2DM: FBG: 152.95 ± 40.95 PBG: 153.07 ± 38.22 HbA1c: 8.12 ± 0.86	- CM: a significant improvement in BG and HbA1c; no significant change in the control group. - In T1DM, the comparison between fasting and postprandial glucose levels was significant in the treatment group, but not the control. HbA1c significant difference in the treatment group. - T2DM: PBG and HbA1c are significantly lower in CM than in the control group.
Agrawal et al., 2011a [33]	India (Bikaner)	24 (70.8)	Age range: 14–16	A randomized, open clinical, parallel, controlled trial	FBG, FI, ID, HbA1c, IA, CP	T1DM	Fresh	500	730 days (>6 months)	FBG: 118.58 ± 19 FI: 18.66 ± 2.81 ID: 32.50 ± 9.99 HbA1c: 7.81 ± 1.39 IA: 19.48 ± 4.22 CP: 0.16 ± 0.08	FBG: 93.16 ± 17.06 FI: 19.05 ± 2.55 ID: 17.50 ± 12.09 HbA1c: 5.44 ± 0.81 IA: 18.76 ± 2.89 CP: 0.21 ± 0.1	FBG: (120.75 ± 17.29 FI: 18.96 ± 3.02 ID: 32.75 ± 11.79 HbA1c: 7.54 ± 1.38 IA: 19.84 ± 3.21 CP: 0.13 ± 0.06	FBG: 122 ± 25.35 FI: 20.84 ± 3.69 ID: 34 ± 10.92 HbA1c: 6.83 ± 1.46 IA: 18.96 ± 2.11 CP: 0.14 ± 0.09	- In the CM group, there were decreases in FBG, HbA1c, and ID.In the control group, there was an increase in insulin requirement. - A significant change in C-peptide levels in both groups as all were on insulin therapy. - No significant changes in FI in CM and control groups. - No significant changes in FI and IA in both groups.
Agrawal et al., 2011b [33]	India (Raica, Non-Raica community)	28 (89)	Age range: 44–54	A crossover study	FBG, FI, HbA1c, PBG, HOMA-IR, CP	T2DM	Fresh	500	90 days (<6 months)	FBG: 184 ± 19 FI: 13 ± 1 HbA1c: 8.4 ± 0.6 PBG: 269 ± 29 HOMA-IR: 6 ± 1 CP: 3.1 ± 0.3	FBG: 161 ± 11 FI: 11 ± 1 HbA1c: 7.3 ± 0.7 PBG: 214 ± 16 HOMA-IR: 5 ± 1 CP: 3.0 ± 0.3	FBG: 86 ± 2 FI: 7 ± 2 HbA1c: 4.9 ± 0.2 PBG: 106 ± 8 HOMA-IR: 1.7 ± 0.5 CP: 1.8 ± 0.2	FBG: 100 ± 3 FI: 10 ± 4 HbA1c: 4.6 ± 0.2 PBG: 97 ± 4 HOMA-IR: 2.5 ± 1.1 CP: 2.5 ± 1.1	- A significant improvement in diabetic and nondiabetic groups when consumed CM. - A significant decrease in FBG. - HbA1c improved after CM consumption. - HOMA-IR decreased in CM.
El-Sayed et al., 2011 [39]	Yemen	45 (66.7)	Age range: 19–20	Randomized study	FBG, HbA1c, PBG, ID	T1DM	Fresh	500	90 day (<6 months)	Group B: FBG: 199.46 ± 4 HbA1c: 9.7 ± 0.39 PBG: 345.6 ± 6.3 ID: 55.1 ± 1.4	Group B: FBG: 155.13 ± 3.5 HbA1c: 7.28 ± 0.23 PBG: 239.2 ± 5.5 ID: 36.2 ± 1.22	FBG: 195.6 ± 2.01 HbA1c: 9.39 ± 0.39 PBG: 339.6 ± 10 ID: 50 ± 0.64	FBG: 173.4 ± 1.66 HbA1c: 9.27 ± 0.36 PBG: 299.1 ± 8.9 ID: 45.46 ± 0.9	- Groups B: a significant decrease in FBG, PBG, and HbA1c - Group A: a significant decrease in FBG and PBG. - Significant decrease in ID in both groups, but % differs.
Mohamad et al., 2009 [42]	Egypt (Cairo)	54 (70.4)	Age range: 17–20	Randomized controlled	FBG, FI, ID, HbA1c, IA, CP	T1DM	Fresh	500	122 days (<6 months)	FBG: 229.9 ± 7.2 FI: 1.83 ± 1.51 ID: 40.83 ± 6.95 HbA1c: 9.459 ± 2.10 IA: 27.92 ± 5.45 CP: 0.22 ± 0.61	FBG: 98.9 ± 16.2 FI: (5.0 ± 1.32 ID: 23 ± 4.05 HbA1c: 7.16 ± 1.84 IA: 20.92 ± 5.45 CP: 2.3 ± 0.51	FBG: 228.2 ± 17.7 FI: 1.75 ± 0.62 ID: 40.83 ± 6.95 HbA1c: 9.59 ± 2.05 IA: 28.20 ± 7.69 CP: 0.24 ± 0.6	FBG: 227.2 ± 17.7 FI: 1.8 ± 0.45 ID: 48.1 ± 6.95 HbA1c: 9.59 ± 2.05 IA: 26.20 ± 7.69 CP: 0.28 ± 0.6	- CM: FBG, IA, and HbA1c are decreased, there are improvements in FI compared with control. - CM: a significant decrease in ID after 16 weeks.
Wang et al., 2009 [45]	China (Beijing)	12 (83.33)	Age range: 49–50	Randomized control trial	FBG, FI	T2DM	Fresh	500	304 days (>6 months)	FBG: 123 ± 19.8 FI: 19.76 ± 2.3	FBG: 94.2 ± 14.3 FI: 6.21 ± 0.56	FBG: 125 ± 18.5 FI: 19.45 ± 2.2	FBG 103 ± 16.7: FI: 7.89 ± 0.67	- CM: significant changes in FBG, FI compared with the start of the study.
Agrawal et al., 2005 [32]	India (Bikaner)	24 (83.3)	Age range: 13–15	Randomized study	FBG, FI, ID, HbA1c, IA, CP	T1DM	Fresh	500	365 days (>6 months)	FBG: 119 ± 19 FI: 6.91 ± 2.13 ID: 32 ± 12 HbA1c: 7.80 ± 1.38 IA: 22.92 ± 5.45 CP: 0.18 ± 0.04	FBG: 95.42 ± 15.70 FI: 18.17 ± 7.12 ID: 17.83 ± 12.40 HbA1c: 6 ± 0.96 IA: 21.84 ± 7.34 CP: 0.24 ± 0.07	FBG: 121 ± 17.3 FI: 7.73 ± 2.42 ID: 33 ± 11 HbA1c: 7.54 ± 1.38 IA: 22.2 ± 7.69 CP: 0.22 ± 0.03	FBG: 105.25 ± 14.50 FI: 19.54 ± 0.43 ID: 30.16 ± 8.45 HbA1c: 7.63 ± 1.03 IA: 19.70 ± 8.40 CP: 0.21 ±0.06	- CM: significant decreases in HbA_1_c, FBG, and ID after treatment compared to before treatment. - No significant change in IA.
Agrawal et al., 2003 [35]	India (Bikaner)	24 (83.3)	Age range: 19–20	Randomized, open case-control, parallel	FBG, FI, ID, HbA1c, CP	T1DM	Fresh ^1^	500	90 days (<6 months)	FBG: 115.16 FI: 16.79 ID: 41.16 HbA1c: 9.54 CP: 1.26	FBG: 100 FI: 16.94 ID: 30 HbA1c: 9.08 CP: 2.22	FBG: 117.16 FI: 16.37 ID: 40 HbA1c: 9.51 CP: 1.24	FBG: 118.16 FI: 16.31 ID: 38.5 HbA1c: 9.48 CP: 2.28	- FBG and HbA1c improved in CM. - No significant difference in FI in CM. - Reduction in the mean doses of insulin in CM.

Glycemic control parameters: Fasting blood glucose (FBG-mg/dL); Postprandial blood glucose (PBG-mg/dL); Glycated hemoglobin (HbA_1_C-%); Fasting serum insulin levels (FI-μlU/mL); Insulin resistance (HOMA-IR); Insulin antibody (IA-%); Insulin dose (ID-U/day); C-Peptide (CP-ng/mL). Blood insulin: 1 μIU/mL = 6.00 pmol/L. Blood glucose: 1 mmol/L = 18 mg/dL 1 Type of DM is not mentioned by authors in this article but was counted as fresh based on previously published studies of the same authors.

**Table 3 nutrients-14-01245-t003:** Subgroup analyses for the different moderators related to the effect of CM intake on two glycemic control parameters (FBG and HbA1c) in patients with diabetes.

Subgroup	Number of Studies	Number of Participants	Effect Estimate [Mean Difference, 95% CI]	*I* ^2^	*p*-Value
Fasting blood glucose levels (mg/dL)					
Type of diabetes					
Type 2 diabetes	8	400	−15.62 [−26.71, −4.54]	66%	0.006
Type 1 diabetes	7	217	−27.20 [−73.97, 19.57]	99%	0.25
Duration of intervention					
>6 months	4	547	−13.26 [−27.38, 0.85]	71%	0.07
≤6 months	11	94	−29.57 [−61.30, 2.15]	99%	0.07
Type of CM					
Pasteurized/Fermented	4	294	1.87 [−2.46, 6.20]	0%	0.40
Fresh	11	347	−30.12 [−61.55, 1.32]	98%	0.06
HbA1c (%)					
Type of diabetes					
Type 2 diabetes	5	368	−1.27 [−2.53, 0.00]	91%	0.05
Type 1 diabetes	7	217	−1.21 [−2.24,−0.19]	92%	0.02
Type of CM					
Pasteurized/Fermented	2	250	−0.31 [−0.45,−0.18]	0.0%	0.00001
Fresh	10	335	−1.50 [−2.26,−0.74]	85%	0.0001
Duration of intervention					
>6 months	3	82	−1.36 [−2.19,−0.53]	71%	0.001
≤6 months	9	503	−1.21 [−2.18,−0.23]	95%	0.02

## Data Availability

Data available on request due to restrictions, e.g., privacy or ethics.

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
