# Peer review of "Effect of Camel Milk on Glucose Homeostasis in Patients with Diabetes: A Systematic Review and Meta-Analysis of Randomized Controlled Trials"

_nutrients, 2022, doi:10.3390/nu14061245_

Round 1

Reviewer 1 Report

The authors have performed a systematic review and meta-analysis of RCT that study the effect of camel milk consumption on diabetes.

The article provides valuable information from different sources gathered together and given significancy.

I only have one comment regarding table 2 which should be reformated as table 1.

Author Response

Reviewer#1

The authors have performed a systematic review and meta-analysis of RCT that study the effect of camel milk consumption on diabetes. The article provides valuable information from different sources gathered together and given significance.

 Comment 1: I only have one comment regarding table 2 which should be reformatted as table 1.

 Response: Done. Table 2 is formatted as Table 1.

 Reviewer#2

This great systematic review with meta-analysis shows the potential of camel milk as a nutritional intervention in diabetes. I only have one real remark:

 Comment 2: It would be nice to add the sixth paragraph of Future Perspectives, proposing how to practically demonstrate the efficacy, efficiency, and effectiveness of camel milk, in both T1DM and T2DM, which could also explain its mechanism of action.

 Response: Thanks for this valuable comment. Added as suggested. Please see lines 261-265.

“Considering the evident mechanisms underpinning the glucose-lowering and the insulin-like effect of CM may prompt clinicians to consider the daily use of two cups of pasteurized CM in patients with T1DM and T2DM as a safe, efficient and effective adjuvant therapy. It is inferred that such adjuvant therapy may reduce the treatment costs and further reduce the dosage of glucose-lowering medications and insulin injections, resulting in less plausible adverse effects”.   

Reviewer 2 Report

This great systematic review with meta-analysis shows the potential of camel milk as a nutritional intervention in diabetes. I only have one real remark:

It would be nice to add a sixth paragraph of Future Perspectives, proposing how to practically demonstrate the efficacy, efficiency and effectiveness of camel milk, in both T1DM and T2DM, which could also explain its mechanism of action.

Author Response

(The authors gave the same response as above.)
